# Liminality: Change Starts Within

Vivianna Rodriguez Carreon 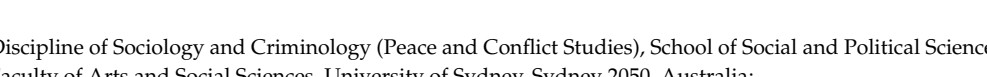

Discipline of Sociology and Criminology (Peace and Conflict Studies), School of Social and Political Science, Faculty of Arts and Social Sciences, University of Sydney, Sydney 2050, Australia; vivianna.rodriguezcarreon@sydney.edu.au or nayamente@gmail.com

**Abstract:** *Change Starts Within* is the welcome title to the Inner Development Goals (IDGs) toolkit. The tools are resources to support the inner growth of individuals and organizations committed to the Sustainable Development Goals (SDGs). This reflection article emerged from reviewing my earlier experiences in inner development while collaborating in the development of the IDGs. Years of continuing inner growth involved going through the liminal stage several times. Evolution is ongoing. Liminality has been conceptualized through different ways of embodying knowing by anthropologists, phenomenologists, psychologists, philosophers, scientists, and spiritual teachers, among others working in transformational processes. Through a lived experience approach, I explore my relationship with the liminal stage. Learning and practicing the "unseen" inner muscle leads to becoming "sensitive" to the subtle qualities. It involves perceiving the world through sensorial qualities, which leads to a conscious action to purposefully commit to what lies along the path to sustainable humanity. At the same time, I notice the limitations for understanding the language of the inner world. The inner world communicates through dynamic manifestations of the lived experience, and when conceptualized in a logical and structured way, it constrains how the animacy of the inner being can be described. The invitation to understand spaces of inner transformation in liminality is to learn to manifest the state of being. The effects of understanding inner development accelerate wisdom and human evolution.

**Keywords:** threshold; Inner Development Goals; liminal stage; inner peace; transformational processes; liminality

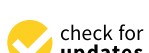



## 1. Introduction

Transformation journeys require what some refer to as going through the threshold [1], a passage between the known and the unknown. Going through this is part of our human evolution process. Rudolf Steiner called it the metamorphosis of the soul, a period made from inner experiences with the outer world [2]. Here, I reflect on inner movements examined through a personal lived experience approach and the meanings of expressing them through different ways of knowing. "As a fundamental human experience, liminality transmits cultural practices, codes, rituals, and meanings in-between aggregate structures and uncertain outcomes" [3]. Uncertainty is a key characteristic of the individual's inner experience in transformation. Moreover, when experienced by whole societies, liminality can serve to identify what Turner explains as periods of crisis, where the future is "unknown", and the situation can dangerously lead to an "institutionalization of liminal conditions" [4]. In this sense, liminality can be defined as the place where collective structures are challenged, transitioning into a new sense of order. In today's landscape, "liminality is a global condition" [5], showing symptoms of the division between self and self (lack of connection to one's own spirituality), self and others (social unrest and collective othering), and self and planet (the inability to sense oneself as part of the Earth) [6]. A division also manifested in language of science, as Robin Wall Kimmerer said "Science can be a language of distance which reduces a being to its working parts; it is a language of objects" [7] (p. 49). However, as this short perspective article discusses, liminality is also

an opportunity to change within [8] and co-create a collective shift in consciousness to integrate the divides.

## 2. The Liminal Period: Learning to See the "Unseen"

Diving into the unknown is not easy; indeed, having a community of support is recommended. For the last few years, as part of my professional development, I have taken several trauma-informed courses led by Thomas Hübl, the author of *Healing Collective Trauma*. In January 2022, it was time to dive in. I joined his two-year intensive program Timeless Wisdom Training (TWT), which the website said was for those "deeply committed to their own personal and spiritual growth" [9]. In the first session, he said, we will learn to walk our talk. This means having the inner tools to be "response-able", being able to respond instead of reacting. For example, when we begin creating inner space in meditation, fragmented experiences or trauma [10], with a big-T or small-t [11], start to emerge as sensations. I learned to stay with and to witness the sensation for what it was, leaving aside the mental context. Explaining personal integration experiences during meditation from the TWT first year in this short paper is challenging. For Hübl, integration means that experiences, frozen in space–time, start melting. By training the inner muscle, these inner abilities enable one "to see" things for what they are and differentiate between one's feelings and those projected from the field. In the liminal space, one is cultivated to respond inwardly.

In oneself, liminality is a period in which we feel there is not much outer action, but sensations manifest and change inside. With practice, manifestations occur when we are in a contemplative state and let the nothing take space within the body. As a result of the inner space made, sensations start to move, and we can begin to recognize whether they are pleasant, unpleasant, or neutral. In TWT, I learned that while it is possible to not sense anything, bringing awareness to the numbness is a recognition of the state of being in the moment. In this transition, we reconnect with our bodies and learn to integrate the sensations as part of us. Some sensations can feel overwhelming for the nervous system, and this is why the threshold is navigated in relation to our higher "I", in support of someone, or in community. When we integrate, we give space to our soul to just be.

## 3. Taking the Step: A Personal Experience

It has been almost two decades since I reflected on my own journey during my fieldwork on the Ayacuchan highlands in Peru when everything seemed to stop for a moment of introspection. Using the words of philosopher Jiddu Krishnamurti, I was freeing myself from what I knew by looking at consciousness without the fragmentation that my mind had created [12]. I realized the answers for my work were lying in the inner space. However, what about my own inner self? As an academic, I intellectually understood the process of introspection, the theories, the approaches, and the strategies. What about the embodiment of knowing, the inner movement that filled me with knowledge and formed my sense of being in the world? While I was accompanying the *campesinos* in the mountains when they were grazing their animals, I sensed inner peace. I felt a sense of totality; yet, no one could see it. Years later, when I was writing about it, I had an *aha* moment.

This is how the *campesinos* build resilience! Pausing to sense awareness enables a sense of inner space where difficult times that arrive can be digested [13].

Yet, how do I prove that the "unseen" is "truth" for others? In 2013, at NSW Parliament House in Sydney, I heard from the 14th Dalai Lama about his late good friend Francisco Varela. The neuroscientist challenged how knowledge is conveyed in a world where most humans rely mainly upon what they can see, when our vision sense does not inform the essential qualities in life [14]. While being in touch with the presence from sensing, the wholeness I experienced was full of knowledge, a different quality than a cognitive one and unseen from the outside. While my journey since then has been gradually developing my inner space, in late 2021, I reflected and made some decisions about the calendar for the year 2022. I wanted to be transformative on the inside, but I did not know what this

meant. Somehow, the words from psychiatrist Bessel Van Der Kolk, whom I had met a decade earlier, reminded me, "In order to change, you need to open yourself to your inner experience" [15].

Before ending 2022, my facilitation colleagues from the Spanish Language Hub of the Presencing Institute (PI) had our last online Zoom meeting of the year. We were having our usual weekly check-in. The proposal question: How was our year in a couple of words? I responded, "in limbo". When trying to explain what I meant, my Argentinian colleague Andrea Fernandez suggested: "You are talking about the liminal stage". In other words, the threshold. It is staying with, being aware of the transition and all the sensations that transformational processes cause. This process of mindfulness and self-awareness is an inner movement, and when being in it requires different qualities to be seen. In an article published in the journal *Nature*, mindfulness and self-awareness was said to lead "to experience a more genuine way of being" [16]. Not judging the sensations that inwardly emerge as "good" or "bad" leads to increasing our awareness to stay with the presence, as well as to stay with the absence. The threshold is where encounters are made. Scientist Stephan Harding said a moment of "encounter" is beyond intellectual processes, where scientific language does not have words to convey [17]. It is the art of being "sensitive" to the unseen. Embodying the qualities to be "response-able", as Hübl said, has a different inner quality than knowing them cognitively.

**4. A Transformational Call:** *The Inner Development Goals* **Framework**

Inner development, individually, collectively, and across systems to build the ability to respond to complexity, is an urgent call. Through some changes that are a natural part of process evolution, the Inner Development Goals (IDGs) framework gradually took form to investigate which skills and capacities are relevant to develop to address sustainability [18]. Between 2020 and 2021, after several initial exploratory conversations and consultations, an initiative was developed to coordinate a process to support the advancement of the United Nations (UN) Sustainable Development Goals (SDGs) [19]. This framework is an umbrella for creating a safe space for people's inner abilities and values to be represented to support their outer sustainability.

Over 1000 people participated in two surveys, and 23 skills were identified. The 23 skills were then grouped into five categories in a co-creative process led by Thomas Jordan of the University of Gothenburg, Sweden. A variety of institutions, organizations, and individuals participated in this process [18] (p.14). The IDGs are organically growing, and the next phase is for other countries to participate and create their 23 skills necessary for their own inner development. The five dimensions are an invitation to experiment and engage in an inner and collective journey [18]: Being (Relationship to Self), Thinking (Cognitive Skills), Relating (Caring for Others and the World), Collaborating (Social Skills), and Acting (Driving Change) [20]. The skills for inner evolution, however, cannot be set in stone. Inner changes occur over time and according to place.

The IDGs project "works to identify, popularize and support the development of relevant abilities, skills and qualities for inner growth" [19] (p. 3). This change, however, is not overnight. It requires embracing the different qualities in which experience manifests and learning to go through several liminal periods. The global symptoms show that "we lack the inner capacity to deal with our increasingly complex environment and challenges" [21]. We need a collective inner shift towards human evolution in wisdom.

At the beginning of 2022, I joined the Scientific Advisory Board for the IDGs. As for many who had been working on the inner space already, the IDGs made complete sense to me from my experience teaching a course on inner and outer peace, where the two forces are interdependent, weaving together. Inner development is necessary for outer development to be conscious and, therefore, sustainable. Stepping into the inner world is challenging for institutions, organizations, and society in general that, for decades, have focused on capitalizing on the outer world. Trusting unseen forces is to be in the liminal period of resolving global problems, and that requires individual and collective evolution.

### 5. Change Starts with the Self: Learning the Language of the Inner World

Transformations require understanding, what Eleanor Rosch called "tuning into" [1]. I will go a step further; it needs sensing the "language" of what we want to understand. It is *iñiy*, a Quechuan word from my native Cusco in Peru. *Iñiy* is a transition state to be able to trust. Like many original oral languages, Quechua is not linear. This word is an expression, a "license" to believe. It does not necessarily mean another individual; for example, in the documentary "My Octopus Teacher" [22], the sense of knowing and interacting is between a human and an octopus. Moreover, it can be the inner movement to trust our own selves when sensing our evolved higher-self version. Lived experiences of sensing are both timelessness and spacelessness. It is in this dynamic of sensing inner space where the inner muscle is built to have the strength to open the door to the unknown. It is the nurture before *Puhpowee*; biologist Robin Wall Kimmerer, a citizen of the Potawatomi Nation located in the USA, explained the meaning in her book *Braiding Sweetgrass*, as "the force which causes mushrooms to push up from the earth overnight" [7]. It is the result of inner development.

Inner growth involves lived experience interacting with sensorial connections, which sometimes awaken unformed new qualities to emerge. For example, when practicing mindfulness meditation and self-awareness, acquiring the sensitivity to step into the threshold is filled with embodiment qualities from a unique connection of sensing the self. For this reflection, mindfulness is "the state of being attentive to and aware of what is taking place in the present" [23], experiencing the state of being without judgment. Meditation is an inner lived experience. Lived experience "provides researchers elements of reflection concerning their *being-in-the-world* as a researcher, their horizon of significance, and their embodiment" [24]. Learning to perceive the world from the body as the place of the senses [25] will lead to a conscious way of doing and accelerate our wisdom in human evolution.

An example of one of the IDGs tools including different ways of knowing is deep listening or *dadirri* as Australian Indigenous Elder Miriam-Rose Ungunmerr Baumann AM said [26]. It is a contemplative practice that can support us when we become companions to ourselves and others while we go through liminality. *Listening to pause* is one of the IDGs tools I had the opportunity to contribute to, in order to support a transition to connection and encounter with our inner selves. Pausing to reflect is embodying what we just heard and developing that inner quality of being more conscious during the listening process to change our response [27]. Understanding ourselves marks the beginning of wisdom [28].

### 6. Conclusions

A call to learn to stay in limbo is training in our unseen inner evolution to gain more wisdom. It is "embracing the mystery and power of transition from what has been to what will be" [29]. In her Ted Talk, Marilyn Bugenhagen calls for all to learn to be liminalists in helping people move through discomfort [30]. However, this cannot happen without embracing the different forms of qualities in which experience manifests and shifting the connoted value given to identities that have been oppressed over centuries. As a paradox, it means giving space for oppressed literacies and identities through enabling the democratization of human expressions. For example, indigenous and ancestral knowledges for centuries suppressed their identities attached to oral languages. Here is the biggest challenge for what we know as knowledge, the attachment to account for what we know. What would have happened if knowledges did not have hierarchical values placed on their forms of expression, and the lived experience of the present in time and space encapsulated the totality of what is known?

The threshold is a passage of life that humans move through several times, individually and collectively. Then, why is liminality, the passage through the unknown, so challenging? This short reflective article has intended to articulate from oral languages and a nonlinear way of constructing knowledge what the challenges are when moving through liminality. From other ways of knowing, inner growth is not meant to be kept as a separate life but

in direct relation to outer development. For decades, structures of power across systems have focused on connecting wellbeing to outer development, living the doing as a separate matter from the being in the inner world. For inner development, as *Puhpowee*, it is working and trusting the unseen forces. Training our "unseen" inner muscle will lead to evolving in our own wisdom. Liminality is the stage of opportunity, where capabilities are transformed to evolve in the individual and collective wisdom as "response-able" humans.

**Funding:** This research received no external funding.

**Data Availability Statement:** The data presented in this article combines different sources, from 3rd party data to personal lived experiences recorded during the author's fieldwork research and inner development work in both personal and professional areas.

**Conflicts of Interest:** The author declares no conflict of interest.

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
