# Peer review of "Liminality: Change Starts Within"

_challenges, doi:10.3390/challe14020025_

Round 1

Reviewer 1 Report

This brief Viewpoint article submitted for consideration in the Special Issue of Challenges on “The Relationship between Sustainability and Inner Development” aims to describe the author’s lived experience of transformational change and inner development through personal study, mindfulness meditation, and professional development, including fieldwork. The story for this Viewpoint is told using the concept of liminality, where the central thesis seems to make the case for a link between inner development, self-awareness, mindfulness, personal transformation, and wisdom. While the central thesis and the core concepts are relevant to the Special Issue, and hence would likely be of interest to the readership and intended target audience, there are a number of significant concerns with writing clarity, content, and structure (i.e., definitions, terminology/jargon, organization, logic/flow, cohesion) that make the story the Viewpoint is trying to tell difficult to follow and hard to comprehend.

My overarching suggestions are to:

(a) Set-up the take-home message of the paper from the beginning (in the Abstract and in the Introduction) by stating a clear, precise, central thesis. Avoid jargon and use simple terms that are easy for a general professional audience to understand. Be sure to clearly define abstract terms, concepts, and/or theories that a general professional audience is unlikely familiar with, like ‘liminality’, ‘inner movement’, ‘embodying knowing’, ‘umbral’, ‘our higher I’, ‘evolved higher self-version’, ‘unseen inner muscles’, ‘outer development’, and ‘better humanity.’ Doing so will aid clarity, which is critical for improving the manuscript so readers are able to comprehend the message.

(b) Restructure and reorganize the paper around the central thesis. Use the Abstract to briefly and concisely tell the reader the whole story. Use the Introduction to introduce (and clearly define) key terms, and tell the reader where you are going, and why. That will help set reader expectations, which along with providing clear definitions of abstract terms/concepts/theories, will aid comprehension. Each section should logically flow from the preceding one, to tell a clear, consistent, cohesive story that, taken together, provides support for the author’s thesis in this Viewpoint.

(c) Whereas a lot of detail and space is devoted to sharing the author’s firsthand experience with personal discovery and inner transformation through work, professional development, and meditation, the connection between Inner Development (and Inner Development Goals, IDGs) and Sustainability (and Sustainability Development Goals, SDGs) is underdeveloped. Given the theme of the Special Issue, which intends to explore the relationship between Inner Development to Sustainability, it would both strengthen the manuscript – and, by extension, bolster fit with the Special Issue – to more fully develop the link between Inner Development and Sustainability, by giving specific, concrete examples based on the author’s lived experience.

(d) Relatedly, I would encourage the author to consult the Special Issue Information posted on the Challenges website (https://www.mdpi.com/journal/challenges/special_issues/15DUC3P984). It would help to make a direct, concrete, explicit connection between the key terms and concepts presented in the Viewpoint with the Keywords and specific intentions described for the Special Issue. Doing so would, again, help make the writing more concrete and understandable to a general audience, while also reinforcing reader comprehension by using consistent core concepts and terminology across the Special Issue topics and the sections within the Viewpoint article.

Finally, given the wide general professional audience that follows work published in Challenges and is likely to see the Special Issue, I would suggest consulting with colleagues from at least 2 different disciplines so they can read the manuscript in the present form, share their own impressions, and give feedback regarding clarity, organization, flow, and comprehension. Colleagues could also consider what I have provided here, in terms of suggestions for aiding clarity and comprehension. It may help if one colleague is steeped in the theories contained in this Viewpoint, since I am not.

Overall, the manuscript lacks a clear central thesis, contains a lot of jargon, and seems to meander as opposed to logically proceeding from one point to another to tell a clear, consistent, cohesive story from start to finish. Clearly stating the thesis, defining abstract terminology, and reorganizing the structure (and key terms/concepts) to parallel the Special Issue topics will make the Viewpoint much less confusing for general readers to follow and easier to understand.

Minor Points

1. The last sentence of the first paragraph sets a reader’s expectation about what the writer will deliver with a piece of scientific prose. In this case, the last sentence in the Introduction is about learning to perceive the world through the body/senses, which leads to conscious doing, and accelerated wisdom and human evolution. Is this the central thesis of the Viewpoint article? If so, then the entire story – from the Abstract, on through all remaining sections – should be used to set-up that central idea, and then explicate it through the various stories of lived experience. Making the abstract more concrete, consistent, and cohesive from start-to-finish will improve clarity and comprehension.

2. In Section 2, the concept of resilience seems to ‘pop-up’ without any context. It’s unclear how exactly resilience fits into the story being told here? What is resilience? Why is it important? How does resilience specifically connect with inner development or sustainability (or the link between those two things, per the Special Issue)?

3. In section 3, the concept of trauma is raised. Like resilience in Section 2, this seems to ‘pop-up’ as a surprise to the reader, because it does not have any set-up or context. Would help to explain how exactly trauma and trauma-informed courses fit into the story being told (central thesis).

4. In section 3, the concept and experience of mindfulness meditation comes up. The widely popular, 8-week Mindfulness-Based Stress Reduction (MBSR) course, developed by Jon Kabat-Zinn and Saki Santorelli, teaches sensory perception and sensory awareness as a new “way of seeing, and way of being”, much like the author describes here. Since many global readers are likely familiar with MBSR if they are interested in mindfulness and meditation, it would help to compare and contrast how popular mindfulness training programs, like MBSR, are similar to or different from the meditation experience, insight, wisdom, and inner/outer transformation being told in this Viewpoint.

5. As noted above, in Section 4 there is not a clear connection made between Inner Development (or IDGs) and Sustainability (or SDGs). It would help to make this connection clear, consistent, and concrete by providing examples of how specific IDGs relate to (or support achieving) SDGs, which is the main focus of the Special Issue.

6. In the Conclusions section, the author refers to organizations, however, the Viewpoint only touches upon organizations and institutions in a very cursory way. If the author wishes to make the direct link between inner development and sustainability vis-à-vis impact on institutional and organizational change, then, again, specific examples are needed to effectively convey that point. And, it should also be tied into the central thesis, as noted, for clarity, consistency, and comprehension.

Author Response

Dear reviewer,

Thank you for your detailed feedback. Your suggestions helped me to make changes and clarify some of the points raised. Here I detailed my responses:

Reply to suggestion a:
Thank you for raising this issue of language. I added important literature about learning the grammar of animacy by Robin Kimmerer and the limitations of language. The inner world contains a whole new language, which Western knowledge is unfamiliar with. 
Regarding what professionals understand, we may have different perceptions of a professional person. Yet, thank you for what was raised. I pointed out the language limitations, and I further explained the concepts.

Reply to suggestion b:
Thank you for this. The abstract and the introduction have been edited and improved by reorganizing around the thesis.

Reply to suggestion c:
Thank you for this point. I created a new section that explains the IDGs specifically. Also, I added points to connect inner development and sustainability. In this section, I provide a summary of the IDGs story. At the same time, I have concepts on inner development and the urgent call for human evolution.

Reply to suggestion d:
Thank you for this. I consulted the Special Issue Information posted. The issue embraces different ways of knowing, including the arts. Here I may not understand your viewpoint of "concrete". Subtle dynamics are specific qualities that require inner knowledge to notice and a new language to communicate. For example, the language of sensitivity enables humans to read and sense qualities emerging in different living organisms, as authors such as Harding, Ungunmerr, and Kimmerer (referenced in the article) said. 

Reply to the final suggestion:
Thank you, the article received 3 other reviewers, where 1 had a minor suggestion, and 2 had very positive reviews.

Replies to minor points raised.

Reply to point 1:
Thank you for these detailed observations. I moved this point to the abstract. Learning to perceive the world through the senses relates to sustainability. 

Reply to point 2:
Thank you, the lived experience of the resilience of the peasants journey attempts to describe the meaning of resilience in that particular context. I added a sentence explaining the relationship with resilience for the rural people.

Reply to point 3:
The concept of trauma has several bibliographic notes. This is a topic that does not just "pop up". It is explained and analyzed thoroughly, particularly with Hubl, Mate, and Van der Kolk conceptualizations.

Reply to point 4:
Thank you for informing me about this particular course. I am familiar with the work of Jon Kabat-Zinn. However, I haven't taken his course. There are several Western and non-western mindfulness courses. The objective of this viewpoint is not to compare programs. My first encounter with meditation, as mentioned, had been contemplation practices done by cultures of oral languages and Indigenous ways of knowing. What is valuable here is the inner language created from practising meditation.

Reply to point 5:
Thank you, this point has been taken, and changes were made to see the connection with sustainability. I added a new section, "A transformational call: The Inner Development Goals framework", and separated it from the personal experience "Taking the step: A personal experience".

Thank you again for taking the time to read and review this short viewpoint article.

Kind regards, Vivianna

Reviewer 2 Report

It's a good study. The section concerning liminality could be a little more developed, namely about the phenomenology of inner change. 

Author Response

Dear reviewer,

Thank you for your feedback. Your suggestion helped me to research further on the literature about liminality and make the connection with the phenomenon of lived experience:

The changes contributed to making a more robust introduction and conclusion. For example, I reflected on the inner movements examined through a personal lived experience approach and the meanings of expressing them through different ways of knowing, including ancestral knowledge. I explained how liminality is a fundamental human experience and transmits cultural practices, codes, rituals, and meanings. At the same time, how liminality relates to individual and collective phenomena.

Thank you again. I am looking forward to seeing published this short viewpoint article. 

Kind regards, Vivianna.

Reviewer 3 Report

The manuscript is modest in its scope but valuable none the less as an informed and articulate personal reflection on inner development related to Sustainable Development Goals.  The concept of liminality as part of inner development is strong.  The manuscript will enrich a wider issue collection of related articles.  Very minor  revisions needed to strengthen the flow of some sentences.

Author Response

Dear reviewer,

Thank you for your feedback and appreciation for its contribution to inner development.

In response to all the reviewer suggestions and your appreciation, the article went through a detailed revision to strengthen the flow of the sentences and the overall coherence and cohesion. 

Thank you again. I am looking forward to seeing published this short viewpoint article. 

Kind regards, Vivianna.

Reviewer 4 Report

Thank you for your paper. I think it will eventually make a contribution after significant modifications. These are the recommendations that I have:

-Start with section four so your reader has a context for this topic

-Carefully walk your reader through the IDGs  and Liminality. Explain liminality and how it is used.

-Provide a more thorough and organized review of the literature on this topic. 

-Use less personal language and comment more on the IDGs. 

-Describe the literature on liminality. This was very confusing.

Again, with further work, the insights in this paper may prove beneficial to others.

Author Response

Dear reviewer,

Thank you for the clarity of your feedback. Your recommendations strengthened the article. Each of the points was actioned to make significant modifications. 

In response to your recommendation, "Start with section four so your reader has a context for this topic", I moved this section after the Introduction. Regarding "Provide a more thorough and organized review of the literature on this topic", I explained the concept of liminality in the Introduction. I reflected on inner movements examined through a personal lived experience approach and the meanings of expressing them through different ways of knowing. I explained how liminality transmits cultural practices, codes, rituals, and meanings. I also connected the concept with liminal conditions in societies and how those are manifested in the collective. The next point raised, "Use less personal language and comment more on the IDGs", helped me to create a new point ", Taking the step: A personal experience". I created another subtitle, "A transformational call: The Inner Development Goals framework", using less personal language and more about the IDGs. I increased and clarified the literature on liminality. The Introduction and Conclusion are connected. The literature on Western inner development and Indigenous ways of knowing intertwined to explain Liminality manifestations in the individual and collective context. The article is now robust with the supported literature.

Thank you again for your specific and detailed points. It has helped strengthen this short viewpoint article and make significant modifications. 

I am looking forward to the insights in this article to benefit others.

Kind regards, Vivianna.

Round 2

Reviewer 4 Report

This paper has significantly improved. The author defines terminology and methods and moves from reporting a personal experience to describing a concept that will benefit others.